# Expanding Horizons: A Review of Sustainability Evaluation Methodologies in the Airport Sector and Beyond

Xibei Jia [1,2,*], Rosário Macário [1,2] and Sven Buyle [1]

1    Department of Transport and Regional Economics, University of Antwerp, Prinsstraat 13,
     2000 Antwerpen, Belgium; rosariomacario@tecnico.ulisboa.pt (R.M.); sven.buyle@uantwerpen.be (S.B.)
2    CERIS, Instituto Superior Técnico, Universidade de Lisboa, Av. Rovisco Pais 1, 1049-001 Lisboa, Portugal
*    Correspondence: xibei.jia@uantwerpen.be

**Abstract:** Addressing a significant gap in the literature, this study commences with a dual focus: assessing sustainability evaluations, both within the airport sector and across a broader range of industries. Through a comprehensive review of 33 academic articles specific to airport sustainability, we delve into a detailed analysis of 16 papers that implement specific methodologies for assessing airport sustainability performance. These methodologies are compartmentalized into three primary categories: Data Envelopment Analysis (DEA) and its extensions, Hybrid Multiple-Criteria Decision Making (MCDM), and composite index-based assessments. A meta-review extending beyond the airport sector uncovers common issues across industries, including the absence of universally adaptable sustainability frameworks and an overemphasis on assessment, overshadowing the essential role of sustainability accounting. Our findings underscore the need for a paradigm shift from pure evaluation towards a holistic approach to sustainability modeling. With systems thinking at its core, this approach allows a better grasp of the complex interactions and feedback loops within sustainability systems and provides a strategy to tackle inherent trade-offs and compensatory effects. By exposing gaps in current practices, this study paves the way for future research, particularly the integration of systems thinking with MCDM, promising to enrich sustainability evaluation and management methodologies, ultimately facilitating more sustainable airport operations.

**Keywords:** airport sustainability; sustainability evaluation; systems thinking





## 1. Introduction

### 1.1. Sustainability and Sustainability Evaluation

Sustainability and sustainable development have been emerging in popularity since the 1990s. Driven by social change, environmental degradation, and the ensuing public interest, sustainability is currently becoming a leading topic in academia, regulators, and business [1]. The most widely accepted definition of sustainable development comes from the Brundtland Commission's 1987 report, *Our Common Future*. The report [2] defines sustainable development as "*development that meets the needs of the present without compromising the ability of future generations to meet their own needs*". In practice, this definition suggests that sustainable development necessitates the adoption of a comprehensive and interconnected approach, which considers the intricate relationships among economic, social, and environmental dimensions. Over the past three decades, the connotations of sustainable development have been enriched with the launch of the Millennium Development Goals (MDGs) [3] and the Sustainable Development Goals (SDGs) [4]. In this process, the quantification of sustainability has also evolved, indicating a growing recognition of the importance of measuring sustainable development progress.

In light of this background, there is a growing tendency for scholars to critically examine and develop alternative methods for evaluating sustainable progress. Current findings suggest that sustainability evaluation has achieved significant breakthroughs in terms

of methodological advancements and the integration of multi-dimensional aspects [5,6]. Despite the burgeoning interest in sustainability evaluation research, several significant limitations persist. To begin with, there is the issue of inconsistencies in both methods and indicators, which often result in incomparable or unreliable outcomes [1,7,8]. Following this, the scope of many studies is frequently narrow and overlooks important social and economic dimensions. This constraint is further compounded by a notable lack of qualitative information integration, which limits the depth of these evaluations [1,5,6,8]. Lastly, there is a dominant short-term perspective in current methodologies that neglects the criticality of long-term sustainability goals, along with an underappreciation of the effects that spatial and situational variations may have on the results [1]. These limitations necessitate further exploration and refinement to improve the robustness and applicability of sustainability evaluation methods in a diverse range of contexts.

Definitions of *"evaluation"* and *"assessment"* can vary among disciplines, and sometimes, these terms are even used interchangeably. However, for the purposes of this paper, we adopt the distinctions made by Scriven [9] and Büyüközkan and Karabulut [1]. According to the widely cited definitions provided in the book *Evaluation Thesaurus* by Scriven [9], *"assessment"* often denotes the process of gathering data, information, or evidence about individuals, groups, or systems, while *"evaluation"* refers to a broader, systematic process that encompasses determining the merit, worth, or value of entities such as programs, policies, or projects. In this sense, assessment can be considered a component of the evaluation process. Within the sustainability domain, Büyüközkan and Karabulut [1] further delineated two essential steps in the evaluation process: *"sustainability accounting"* and *"sustainability assessment"*. The *"accounting"* aspect is closely associated with determining which information to collect for specific purposes, defining appropriate indicators, and measuring them, necessitating robust conceptual models such as indicator sets. The *"assessment"* aspect involves assigning meaning to the collected qualitative and quantitative data through analytical integration techniques. Once accounted for and assessed, the overall sustainability performance can be reported as a strategic tool for corporate management and communication. We recognize that the use and interpretation of these terms can vary among researchers and that our usage here is informed by the sources cited and the specific context of our study.

In summary, this paper recognizes sustainability evaluation as a process encompassing both sustainability accounting and sustainability assessment, where accounting represents the selection and measurement of appropriate indicators, while assessment involves the application of analytical methods to assign meaning to the collected data, ultimately providing an integrated understanding of an entity's overall sustainability performance.

*1.2. Airport Sustainability Evaluation*

Serving as integral hubs in the air transportation network, airports hold a vital position in the pursuit of sustainable development. While air transport results in a range of negative environmental consequences, such as air and noise pollution, it also provides a plethora of benefits. These benefits encompass economic growth, including enabling worldwide trade, enhanced connectivity, and the creation of job opportunities, among others [10]. In this context, the emphasis on sustainability becomes increasingly essential, as it seeks to integrate environmental, social, and economic considerations into airport planning, operations, and development, with the goal of minimizing negative impacts and maximizing benefits. By adopting and implementing sustainable practices and strategies, the aviation sector can effectively navigate the delicate balance between environmental concerns and socio-economic progress [5].

Airport sustainability evaluations are therefore essential for identifying areas for improvement and monitoring progress over time to ensure that sustainability goals and targets are met. These evaluations provide valuable insights for decision-making and facilitate informed choices about resource allocation, policy and investment priorities. In addition, evaluations allow for benchmarking against established sustainability indicators,

facilitate performance comparisons across airports, and promote knowledge sharing and industry-wide improvements [11]. Evaluations can also stimulate innovation by identifying sustainability gaps, increase transparency and accountability among airport operators, regulators, and stakeholders, and encourage stakeholder engagement and collaboration. Meanwhile, demonstrating a commitment to sustainability can enhance an airport's reputation and competitiveness, attracting passengers, airlines and businesses that prioritize sustainability [12,13].

Despite the myriad benefits and essential nature of airport sustainability evaluations, and the efforts made by the industry to align with sustainable practices, a notable gap exists in the academic literature. To date, no comprehensive review of airport sustainability evaluation studies has been conducted, leaving existing assessment methodologies unexamined. This oversight raises three pertinent questions that remain unsolved:

- What are the targets and focuses of current airport sustainability evaluation studies?
- Through which methodological approaches is airport sustainability assessed?
- Which limitations are inherent in the current methodologies, and how might they be effectively mitigated or addressed?

To tackle these three research questions from a holistic perspective, this paper employs a systematic literature review as its primary research methodology. Additionally, considering the small pool of research dedicated to airport sustainability evaluation, this paper will perform two sorts of reviews based on the concept of *"looking inside the box"* and *"thinking outside the box"*. The former, *"looking inside the box"*, refers to examining studies centered on sustainability assessments of airports. The latter, *"thinking outside the box"*, entails drawing comparisons between airport sustainability evaluation studies with those conducted in other sectors. The analysis of studies beyond the airport sector will be executed using a *"meta-review"* approach, signifying a review of existing literature review papers that examine sustainability evaluation studies in a wider context.

The remainder of this paper is organized as follows: Section 2 delineates the methods and materials employed in conducting the systematic literature review; Section 3 presents the findings pertaining to airport sustainability evaluation, Section 4 offers a comparative analysis between airport-specific studies and those conducted in sectors beyond airports; finally, Section 5 provides a comprehensive conclusion, synthesizing the key insights and implications derived from the analysis.

## 2. Methods, Materials and Content

As the initial endeavor to synthesize a comprehensive understanding of current airport sustainability evaluation studies, this paper utilizes the systematic literature review method. The research process is depicted in Figure 1, illustrating a three-stage procedure consisting of literature identification, filtering, and inclusion.

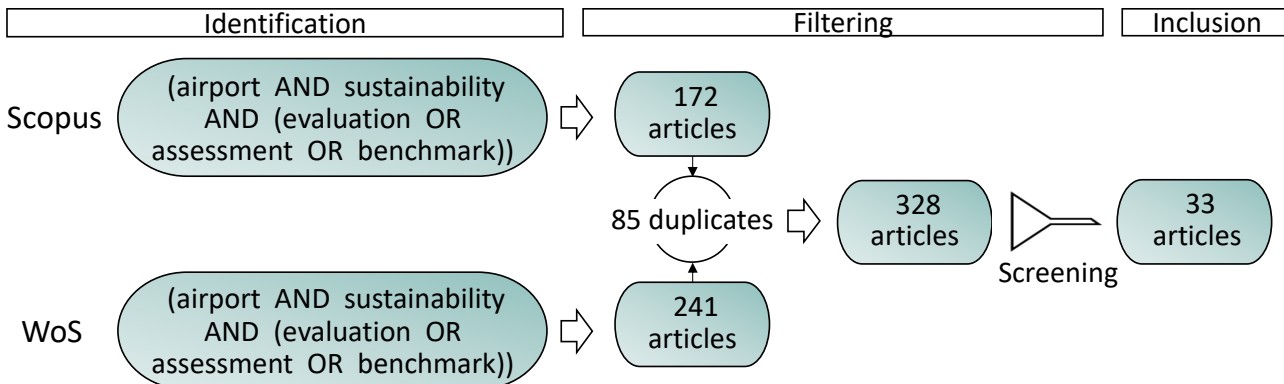

**Figure 1.** Flowchart of systematic literature review.

In the identification stage, Scopus and Web of Science (WoS) are selected as the primary databases for conducting the literature search, given their status as two of the most extensive and comprehensive sources of publication metadata and impact indicators available [14]. As preeminent databases in the academic sphere, both Scopus and WoS provide access to a substantial collection of peer-reviewed articles, ensuring a thorough and rigorous search for relevant literature pertaining to airport sustainability evaluation studies. The search is conducted using the following keywords: (airport AND sustainability AND (evaluation OR assessment OR benchmark)), encompassing all available publications until January 2023. This process yielded 172 articles from Scopus and 241 from WoS.

Subsequently, the filtering process commences, with duplicate checking across both databases. Upon excluding 85 duplicate papers, the remaining pool consists of 328 articles. The second round of filtering is conducted to refine the selection further based on two specific criteria: (1) inaccessible information or language constraints, wherein the full paper is unavailable, or the paper is not written in English, and (2) deviation from the research scope, where the research target is not an airport, or the focus is not on airport sustainability. The filtering stage ensures the inclusion of relevant and accessible papers for an in-depth analysis of airport sustainability evaluation studies.

A total of 33 papers were ultimately selected for our analysis. Upon conducting a thorough review, it became evident that, although all papers pertain to airport sustainability evaluation, their content varies considerably. The 33 studies can therefore be classified into four distinct categories based on their evaluation content, with their distribution illustrated in Figure 2.

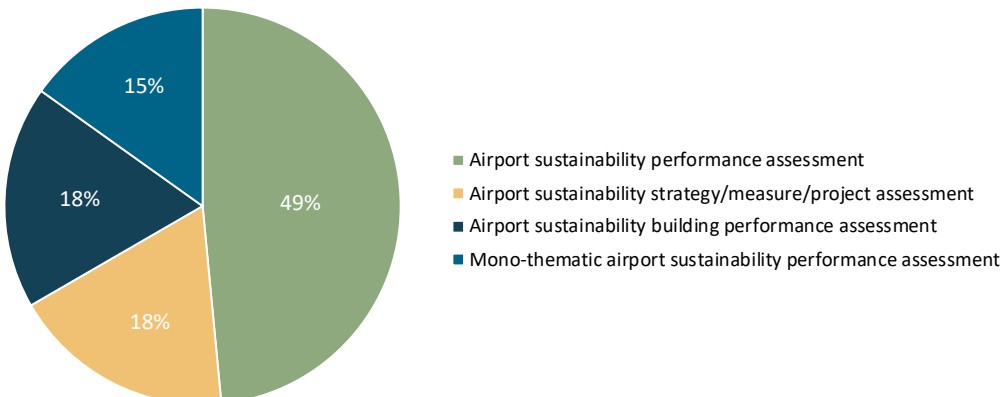

**Figure 2.** Content distribution of airport sustainability evaluation studies.

- **Airport sustainability performance evaluation studies**: These studies focus on quantifying an airport's overall sustainability using performance indicators, with an emphasis on the operational phase. This category constitutes more than half of the analyzed airport sustainability evaluation studies, reflecting the diversity of methods employed. A more detailed discussion of this type of study will be provided in Section 3.
- **Airport sustainability strategy/measure/project evaluation studies**: This category involves quantifying the sustainability impact (either positive or negative) of a specific strategy, measure or project on the airport. Of the six papers, two are review papers [15,16], and two concentrate exclusively on sustainability accounting [17,18]. The remaining two papers integrate both sustainability accounting and assessment. For instance, Dimitriou and Karagkouni [11] applied comprehensive performance benchmarking to assess the sustainability of environmental mitigation strategies at 20 airports in the United States, Asia, and Europe. Similarly, Li and Loo [19] utilized Cost-Benefit Analysis (CBA) to evaluate the sustainability of two airport infrastructure projects at Hong Kong International Airport.
- **Airport terminal sustainability evaluation studies**: These articles systematically analyze and evaluate airport terminals' design, construction, operation, and maintenance, aiming to ensure sustainable performance throughout their life cycle [20–25]. Green

building rating tools (GBRT) have been identified as the primary method for conducting such research. The most recent articles also revealed hybrid solutions combining GBRT with methods such as the Classification and Regression Tree (CART) model [24] and Life Cycle Analysis (LCA) [21].

- **Mono-thematic airport sustainability evaluation studies**: These studies are focused on assessing the sustainability of specific subsystems or themes within the entire airport operation. Rather than looking at a specific structure or facility, these studies delve into particular operational areas such as the water management system [26,27], waste management system [28], energy management system [29], and pavement system [30]. Given the distinctive characteristics of each theme, the evaluation methods used in those studies exhibit remarkable diversity and adaptability in addressing the particular challenges associated with each theme.

In pursuit of a cross-disciplinary comparison, as stated in the introductory section, this paper will also conduct a meta-review of sustainability evaluation studies, with a particular emphasis on holistic performance evaluations in sectors beyond airports. Utilizing a comparable literature identification, filtering, and inclusion process, we ultimately include seven review papers for further analysis. A synopsis of these papers is provided in Table 1:

**Table 1.** Synopsis of sustainability performance evaluation review papers.

| No. | Citation | Review Period and Quantity | Scope of Review |
|---|---|---|---|
| 1 | Walzberg et al. [31] | Not mentioned | Methods for sustainability performance evaluation in Circular Economy (CE) |
| 2 | Turkson et al. [6] | Not mentioned | Methods for sustainability performance evaluation in energy production system |
| 3 | Büyüközkan and Karabulut [1] | 2007–2018 128 | Methods for sustainability performance evaluation (covering all sectors) |
| 4 | Bueno et al. [7] | Not mentioned | Tools and methods for sustainability performance evaluation of transport infrastructure |
| 5 | Ghadimi et al. [5] | 1987–2012 111 | Tools and methods for sustainability performance evaluation of manufactured product and manufacturing process |
| 6 | Singh et al. [32] | 1998–2018 128 | Sustainable indices applied in policy practice (covering all sectors) |
| 7 | Gasparatos et al. [8] | 2000–2020 92 | Methods for sustainability performance evaluation (covering all sectors) |

## 3. Sustainability Performance Evaluation within the Airport Sector

In this Section, we will delve further into studies that aim to evaluate airports' sustainability performance, with a primary focus on two critical aspects. Firstly, we examine the theoretical basis of sustainability accounting, which involves the selection of appropriate sustainability indicators. Secondly, we explore the methods employed for sustainability assessment, seeking to identify the effectiveness and limitations of existing approaches. A comprehensive overview of airport sustainability performance evaluation studies is provided in Table 2.

### 3.1. Descriptive Analysis

As shown in Table 2, there are three papers focused solely on airport sustainability accounting, while 12 papers incorporate both accounting and assessment, and one paper serves as a review. Airport sustainability was first accounted for by Upham [33] in 2001 in his work titled *Selecting indicators for a decision support tool for airport sustainability*. Upham proposed an airport sustainability model consisting of nine indicators: number of surface access vehicles, aircraft movements, static power consumption, gaseous pollutant emissions, aircraft noise emissions, terminal passengers, surface access passengers, water

consumption, and solid waste. In 2015, Upham and Mills [34] expanded the framework to include land take and biodiversity. According to both of Upham's articles, airport sustainability encompasses two dimensions: environmental and operational. While these works provide valuable insights into selecting sustainability indicators for airport sustainability accounting, they lack a clear, transparent, and well-defined theoretical framework to guide and inform the indicator selection process.

In 2010, Janic [35] laid the groundwork for accounting airport sustainability through the concept of *"airport as a system"* and the theory of *"effects–benefits"* and *"impact-externalities"* associated with airport activities. The *"airport as a system"* emphasizes the interconnectedness of different components within the airport and its broader context, recognizing that changes in one subsystem can have impacts on others. Meanwhile, *"effects–benefits"* and *"impact–externalities"* refer to the positive outcomes and negative impacts, respectively, of airport activities. To account for airport sustainability, Janic proposed an indicator system comprising 12 indicators: four for operational (demand, capacity, quality of service, and integrated intermodal service), two for economic (profitability and labor productivity), one for social (direct and indirect employment), and five for environmental performance (energy efficiency, noise, air pollution, land use efficiency, and waste efficiency). Janic's work represents a significant theoretical revolution in airport sustainability accounting, providing a useful foundation for understanding the interrelationships between different components of the airport system. However, a limitation of the framework is that it does not explicitly address the interconnections between the 12 indicators or the potential impact of changes in one subsystem on others. However, prior to 2010, there was no established method for effectively integrating the sustainability indicators used in airport sustainability accounting.

Adler et al. [36] benchmarked the performance of 85 European regional airports through Data Envelopment Analysis (DEA). Although the authors labeled their work as sustainability performance benchmarking, the selected input and output indicators do not differ from those previously employed in DEA for assessing airport operational efficiency. The authors attribute this to the financial and operational constraints faced by small regional airports, which can make achieving sustainability a challenge. In 2016, Kılkış and Kılkış [37] introduced a composite indicator approach for benchmarking airport sustainability performance, encompassing 25 indicators across five dimensions, with a particular focus on environmental sustainability. The study demonstrated the feasibility of using the composite indicator approach for evaluating airport sustainability performance. Furthermore, the authors explored the implications of their findings for airport management and policy, classifying measures into categories that airports can control, guide, or influence.

**Table 2.** An overview of airport sustainability performance evaluation studies.

| No. | Citation | Type of Sustainability Study | | | Theoritical Basis for Accounting | Accounted Dimensions * | | | | No. of Indicators | Assessment Methods | Case Study |
|---|---|---|---|---|---|---|---|---|---|---|---|---|
| | | Accounting | Accounting and Assessment | Review Paper | | Ec | En | O | S | | | |
| 1 | Dimitriou and Karagkouni [11] | | X | | 1. literature review | | X | | | 16 | Linear scoring method | Top 5 regional tourist airports in Mediterranean islands |
| 2 | Yangmin et al. [38] | | X | | 1. Literature review 2. Airport-Industry-City (AIC)'s synergy and sustainability development | X | X | X | X | 18 | Synergetic Measure Model (SMM) | Zhengzhou international airport |
| 3 | Kucukvar et al. [39] | | X | | 1. Literature review | X | X | X | X | 8 | Data Envelopment Analysis (DEA) | 30 major international airports |
| 4 | Kumar et al. [40] | | X | | 1. Literature review 2. Delphi method, and panel discussion with experts | | X | X | X | 43 | Best worst method (BWM) and VIKOR method | 5 Indian airports |
| 5 | Kaya and Erginel [41] | | X | | 1. Literature review 2. Brainstorming of experts | | X | | X | 15 | Stepwise Weight Assessment Ratio Analysis (SWARA) | Ankara Esenboga Airport |
| 6 | Greer et al. [42] | | | X | | | | Not applicable | | | | |
| 7 | Wang and Song [43] | | X | | 1. Literature review | X | | X | | 7 | DEA | 8 Chinese airports and 4 Asian airports |
| 8 | Wan et al. [44] | | X | | 1. Literature review | X | X | X | X | 55 | Synthetic evaluation method | Guangzhou Baiyun International Airport |
| 9 | Lu et al. [45] | | X | | 1. Literature review 2. Experts' interviews, and brain storming | X | X | | X | 15 | Hybrid Multiple-criteria decision making (MCDM) | 3 Taiwanese airports |
| 10 | Carlucci et al. [46] | | X | | 1. Literature review | X | | X | | 9 | DEA | 34 Italian airports |
| 11 | Olfat et al. [47] | | X | | 1. Literature review | X | X | X | X | 9 | Fuzzy dynamic network DEA | 28 Iranian airports |
| 12 | Kılkış and Kılkış [37] | | X | | 1. Literature review 2. Consultation with experts | | X | X | | 25 | Sustainability ranking index | 9 world busiest and best airports |
| 13 | Adler et al. [36] | | X | | 1. Literature review | X | | X | | 7 | DEA | 85 European regional airports |

**Table 2.** *Cont.*

| No. | Citation | Type of Sustainability Study | | | Theoritical Basis for Accounting | Accounted Dimensions * | | | | No. of Indicators | Assessment Methods | Case Study |
| --- | --- | --- | --- | --- | --- | --- | --- | --- | --- | --- | --- | --- |
| | | Accounting | Accounting and Assessment | Review Paper | | Ec | En | O | S | | | |
| 14 | Janic [35] | X | | | 1. Literature review 2. *Effects-benefits and impacts-costs* theory | X | X | X | X | 12 | n.a. | n.a. |
| 15 | Upham and Mills [34] | X | | | 1. Literature review | | X | X | | 10 | n.a. | n.a. |
| 16 | Upham [33] | X | | | 1. Literature review | | X | X | | 9 | n.a. | n.a. |

*: Ec: Economic; En: Environmental; O: Operational; S: Social. (In the context of this table, the *"Operational"* dimension pertains to the internal performance metrics of an airport, such as aircraft movements and passenger or cargo throughput. This dimension is focused on the efficiency and effectiveness of operations, while sustainability implications of these operations are captured under the respective Economic, Environmental, and Social dimensions).

Despite the careful selection of indicators through an extensive literature review and consultation with experts in the field of airport sustainability, the study still exhibits a degree of bias resulting from the subjectivity of the assigned weightings. The authors assigned equal weight to all indicators, which may not accurately reflect the relative importance of each sustainability aspect. In the same year, Olfat et al. [47] employed a Fuzzy dynamic network DEA to evaluate the sustainability performance of 28 Iranian airports. Unlike traditional DEA, which treats the decision-making process as a *"black box"* and considers only primary input and output values, the network DEA enables a more comprehensive analysis of airport sustainability performance by accounting for the interactions between the airport, the community, and passengers. This approach aligns with the *"airport as a system"* concept discussed earlier. However, the model has limitations, such as the subjectivity of weights assigned to different indicators and the challenge of quantifying fuzzy concepts, such as service quality and satisfaction. Furthermore, the model does not explicitly incorporate feedback loops and causal relationships or consider the broader system in which airports operate, including the global aviation industry and the overarching economic, social, and environmental contexts.

In the succeeding years, DEA and its extensions have gained popularity for assessing airport sustainability. Carlucci et al. [46] applied the classical form of DEA to evaluate the efficiency and economic sustainability of 34 Italian regional airports. Wang and Song [43] utilized DEA to assess the sustainability performance of 12 Asian airports; however, despite their claim of using a *"network DEA"* approach, their methodology more closely resembled a traditional DEA model. While their study incorporated input (runway area and passenger terminal area), intermediate (processed passengers, processed cargo, and aircraft movements), and output (airport total revenues and airport net income) variables, it did not explicitly model the interdependencies and interactions between these variables, which is a key characteristic of network DEA. Kucukvar et al. [39] constructed four different DEA models using input-oriented modeling with multiple undesirable environmental inputs (energy, carbon, water, and waste) and desirable outputs (revenue, passengers, and employment) to compare sustainability performance levels of 30 major international airports in various contexts.

Hybrid Multiple-Criteria Decision Making Analysis (MCDM) was first introduced for airport sustainability performance assessment by Lu et al. [45]. The authors employed a three-step process, starting with DEMATEL (Decision-Making Trial and Evaluation Laboratory) to establish an influential-network-relationship-map. Subsequently, they used DANP (DEMATEL-based Analytical Network Process) to determine influential weights, followed by a hybrid modified VIKOR for selecting and improving performance gaps between aspiration values and the current situation of the international airport. Although the DEMATEL method effectively identifies cause-and-effect relationships among factors and evaluates their mutual influence, it does not probe the underlying reasons or mechanisms behind these influences, necessitating further investigation through complementary research methods. In the article, sustainability factors were derived solely through a literature review and the opinions of 15 experts, which introduces a significant degree of subjectivity. Moreover, the degree of influence could vary depending on the operational environment and focus of the airports. A more methodical approach would initially entail delineating the relationships among the factors, followed by a quantification of their respective influences. In other words, a sound theory for sustainability accounting is the basis for the subsequent sustainability accounting. Despite these limitations, as a pioneering work in applying hybrid MCDM to airport sustainability assessment, this paper offers valuable insights into quantifying interactions between sustainability criteria through participatory methods. Kaya and Erginel [41] employed the Stepwise Weight Assessment Ratio Analysis (SWARA) and Hesitant Fuzzy Sustainable Quality Function Deployment (HFSQFD) methods to evaluate the sustainability performance of Ankara Esenboga Airport, while Kumar et al. [40] utilized the Best worst method (BWM) and VIKOR methods to assess the sustainability performance of five Indian airports. Although these MCDM-based

sustainability performance assessment studies differ in the selection and combination of MCDM techniques, they share the same limitations as that by Lu et al. [45].

Wan et al. [44] applied the composite indicator approach for the second time in airport sustainability assessment, utilizing the Min-max method for data normalization and Benefit-of-the-Doubt (BoD) for data weighting. Despite incorporating 55 indicators spanning economic, environmental, social, and operational dimensions, trade-offs and subjectivity remain inherent in the study. Furthermore, the authors selected indicators solely based on literature review, overlooking potential overlaps between different indicators and dimensions. Similar shortcomings were identified in the research by Dimitriou and Karagkouni [11], who employed the linear scoring method to evaluate the top five regional tourist airports in Mediterranean islands.

In our literature review, the sole review paper identified is from Greer et al. [42], which specifically centers on environmental sustainability metrics and methods for airports. The review encompasses an analysis of 108 articles and technical reports, emphasizing the need for systematic assessment that considers diverse emissions and regional variations. The paper also highlights the significance of stakeholder involvement, life-cycle assessments, and establishing connections between environmental impacts and operational outcomes in future research. Additionally, it underscores the importance of addressing global challenges such as resilience, climate change adaptation, and mitigation of infectious diseases.

Yangmin et al. [38] made significant progress in airport sustainability by employing the Synergetic Measure Model (SMM) approach to assess the synergy and sustainability of the Airport–Industry–City (AIC) system in Zhengzhou Aerotropolis in China. The study reinforces the *"airport as a system"* principle, acknowledging that an airport cannot achieve sustainability in isolation. Therefore, the scope of the study was set as *"Aerotropolis"*. The sustainable association of AIC consists of internal core system associations and external system associations. The internal core system connection entails the synergetic relationship between the airport, industry, and city, while the external relationship of AIC concerns the association of sustainable development with the economy, society, and ecosystem. These subsystems and the AIC system interact and mutually reinforce each other, promoting industrialization, commercialization, urbanization, population growth, and green development. AIC represents a dynamic, diverse, and open complex system, with driving factors potentially originating from the interaction of the system's elements or from governmental guidance based on specific strategic goals. Upon constructing the AIC sustainable mechanism theory, Yangmin et al. [38] selected positive and negative outward indicators to create a synergetic evaluation index system.

### 3.2. Assessment Methodologies

In the previous Section, we reviewed 16 studies that focused on airport sustainability performance evaluation. Although those studies employed a range of methodologies, they can be categorized into three types, as outlined in Figure 3: DEA and its extensions, Hybrid-MCDM, and composite index-based assessment. Based on the taxonomy of methodologies, this Section seeks to elucidate the key findings in relation to airport sustainability evaluation.

(1)　Fuzzy Dynamic Network DEA excels as a DEA tool for airport sustainability evaluation, yet enhancing its accuracy demands reduced subjectivity and integration of feedback loops and causal relationships in the sustainability system.

In its classic form, DEA focuses primarily on operational efficiency and does not take into account external factors generated by airport operations, such as noise pollution, air emissions and waste generation. As a result, it lacks a holistic perspective from which to address sustainability. Furthermore, the classic form is subject to the *"curse of dimensionality"* [48,49], which refers to the decrease in discrimination power as the number of inputs and outputs increases relative to the Decision Making Units (DMUs), potentially leading to a higher proportion of DMUs being identified as efficient, even if they are not. As a static analysis method, the classic DEA also considers inputs and outputs as independent

entities, ignoring interdependencies and the dynamic nature of operations. Additionally, the method does not account for variability, imprecision, or vagueness in input and output data. To overcome these limitations, researchers have developed new methods, such as Network DEA and Fuzzy Dynamic Network DEA. The Network DEA divides the airport's overall sustainability into interconnected stages or sub-processes, each with its own inputs, intermediates, outputs, and performance indicators, considering the interdependencies within a sustainability system. Meanwhile, the Fuzzy Dynamic Network DEA uses fuzzy logic to handle imprecise or uncertain data and capture the dynamic nature of airport sustainability over time by considering the inter-temporal relationships between inputs, outputs, and intermediate products. However, both methods do not inherently consider feedback loops and causal relationships among the inputs and outputs or within subsystems. Gaining an awareness of these relationships enables decision-makers to identify potential unintended consequences, trade-offs, and synergies, ultimately fostering more informed and holistic strategies for enhancing airport sustainability performance.

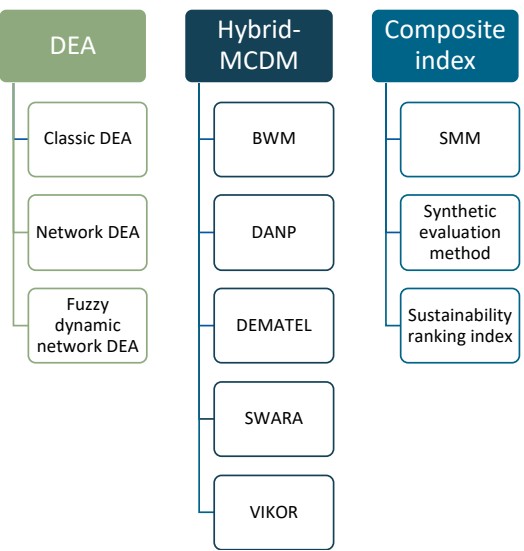

**Figure 3.** Taxonomy of airport sustainability performance assessment methodologies.

(2) Hybrid-MCDM constitutes a useful complementary instrument for evaluating airport sustainability; however, it is inadequate to serve as the predominant methodology.

As a decision-making facilitation tool, MCDM assists decision-makers in evaluating and ranking alternatives based on multiple criteria, accounting for trade-offs, and pinpointing priorities for enhancement. Nevertheless, MCDM alone is insufficient to address the intricate relationships within an airport sustainability system, attributable to three factors: limited scope, static nature, and subjectivity. Firstly, Hybrid-MCDM methods primarily focus on appraising and ranking alternatives using multiple criteria, rather than modeling the underlying structure and dynamics of a complex system, such as airport sustainability, which encompasses numerous interlinked components and processes. Secondly, MCDM methods are generally static, concentrating on a specific point in time, failing to capture the dynamic nature of airport sustainability systems, which involve evolving conditions, interactions, and performance over time. Lastly, MCDM methods often depend on subjective judgments and expert opinions, potentially introducing biases and uncertainty into the analysis. This subjectivity may constrain the capacity of Hybrid-MCDM to deliver an objective understanding of the airport sustainability. However, taken as a supplementary tool, hybrid-MCDM offers multiple perspectives, identifies trade-offs and priorities, encourages stakeholder participation, and exhibits flexibility. When used in combination with methodologies that capture system dynamics and interdependencies, it can promote a more holistic understanding of airport sustainability and help make informed decisions.

(3)   The SMM offers a more advanced perspective on airport sustainability evaluation compared to the composite indicator approach, but experts' consultation and stakeholders' engagement need to be incorporated to form a robust systems-oriented view.

The SMM shares similarities with the composite indicator approach, particularly in the calculation process, as both aggregate multiple indicators to derive a single, quantifiable measure for complex systems. While composite indicators typically prioritize measuring overall performance or effectiveness without explicitly addressing synergy, SMM specifically focuses on quantifying synergy within a complex system, which encompasses the degree of cooperation and coordination among various components or subsystems. In SMM, selecting order parameters is of paramount importance, as they must accurately represent such cooperation and coordination. In contrast, composite indicators may employ a broader range of indicators that do not directly capture synergistic interactions within a system. This leads the SMM to reveal a more advanced approach to airport sustainability evaluation compared to the composite indicator approach in the paper pool of this literature review. However, Yangmin et al. [38] selected order parameters solely based on a literature review, which introduced a high degree of subjectivity and uncertainty. To address this limitation, expert consultation and stakeholder input should be incorporated. In addition, in the paper by Yangmin et al. [38], the main focus is on quantifying synergies and capturing interactions, without paying adequate attention to the validity, correlation, or robustness of individual indicators. As these indicators play a pivotal role in shaping the understanding and composition of the system, their rigorous examination is paramount. Furthermore, the application of sensitivity analysis could significantly contribute to reinforcing the model's reliability and robustness.

Each of the aforementioned methods offers unique strengths and limitations in assessing airport sustainability. Although some methods excel in specific aspects, no single approach is flawless or fully captures the complexities and nuances inherent in airport sustainability evaluations. Combining different methodologies and capitalizing on their respective strengths holds the potential to yield a more comprehensive and integrated assessment of airport sustainability. By integrating various approaches, researchers can address individual methodological limitations and better understand the complex relationships and dynamics within the sustainability system.

Moreover, owing to the limited number of studies on airport sustainability evaluation, it is crucial to examine sustainability evaluation methodologies in sectors beyond airports. Exploring the methods utilized by other industries could uncover innovative and transferable strategies suitable for airport assessments. Furthermore, analyzing the similarities and differences between airport sustainability evaluations and those conducted in other sectors can provide valuable insights and contribute to the development of more robust evaluation frameworks.

## 4. Comparative Discussion beyond Airport Sector

Upon examining the review papers on sustainability evaluation methodologies outlined in Section 2, we will engage in a comparative analysis in this section. The following conclusions can be drawn from such comparison of sustainability evaluation methodologies used by the airport sector and beyond:

(1)   The majority of sustainability evaluation papers devise their own approach and criteria, indicating the lack of universally adaptable, well-defined sustainability frameworks.

In the airport sustainability performance evaluation studies cataloged in Table 2, each deploys unique criteria and approaches rather than directly adapting existing frameworks. This phenomenon is aligned with the findings from Büyüközkan and Karabulut's [1] cross-sectoral review of sustainability performance evaluation methodologies. Their analysis of 111 papers revealed that 80% devised their own frameworks, while the remainder employed pre-existing ones. This observation accentuates a prevalent trend in analytical literature of originating distinct sustainability hierarchies and attributes, as opposed to applying established evaluation models. Several plausible explanations exist for this practice. First,

requisite data for evaluation may be unavailable, necessitating the creation of fresh, suitable criteria. Second, criteria may be too subjective for specific applications, leading to a unique development each time. Third, researchers may have different interpretations of terminology and, as our collective understanding of sustainability evolves, researchers may find existing frameworks to be outdated or incomplete.

This trend underscores the need for future research to focus on the development of universally adaptable, comprehensive sustainability frameworks that can be readily applied across different airports, thereby enhancing comparability and consistency in sustainability evaluations.

(2) The preponderance of sustainability evaluation studies predominantly emphasizes the assessment aspect, often neglecting the foundational concept of sustainability accounting.

At the outset of this paper, we established that a comprehensive sustainability evaluation should incorporate two critical elements: accounting and assessment. However, as delineated in Table 2, the majority of airport sustainability evaluations currently lack a cohesive theoretical framework to support their accounting, relying instead on literature reviews or participatory methods. Only two papers—one by Janic [35] proposing the *"Effects–benefits and impacts–costs"* theory, and another by Yangmin et al. [38] outlining the AIC's synergy model—attempted to construct a conceptual theory for sustainability accounting. This trend is not isolated to the airport sector, as corroborated by Singh et al. [32] and Büyüközkan and Karabulut [1], highlighting a widespread issue within sustainability evaluations across sectors. Apart from the inherent complexity and evolving nature of the concept of sustainability, the relative neglect of the accounting aspect can also be attributed to its inability to yield immediate tangible results that contribute to direct and actionable conclusions. However, in the absence of a comprehensive sustainability accounting foundation, ensuing assessments risk being skewed, incomplete, or misleading, thus potentially compromising the effectiveness of sustainability initiatives. Consequently, irrespective of its less immediate or tangible results, the significance of sustainability accounting within the entirety of the evaluation process remains paramount.

(3) Certain scholars beyond the airport sector have argued that the traditional understanding of sustainability (which typically segments it into three or four separate dimensions) is losing ground.

In both the accounting and assessment phases within the airport sector, the three-pillar model, encompassing economic, environmental, and social dimensions, or alternatively, the four-pillar model, which includes an additional operational dimension, is commonly employed, particularly in indicator categorisation. This categorisation method is widely observed across various sectors, with scholars striving to balance the selection of sustainability indicators [1,5,7,8,32]. Büyüközkan and Karabulut [1], suggesting that the traditional conception of sustainability, which prescribes a clear distinction among sustainability dimensions, is gradually being undermined. The demarcation lines among the three (or four) sustainability pillars can blur due to mutual dependencies, subjective interpretations of criteria, and overlapping impacts, thereby complicating the neat allocation of each indicator to a singular pillar. For instance, consider airport noise. While it can be classified under the environmental dimension as a form of pollution, noise mitigation also falls within the social responsibilities of the airport operator. The number of noise-related complaints can provide a measure of such social responsibility. In such scenarios, traditional understanding may not adequately explain the role of noise in airport sustainability. On the other hand, the blurring boundaries between sustainability dimensions challenge the traditional decision-making and prioritization processes. It necessitates an integrated approach recognizing the interdependence of pillars, as decisions in one area can have ripple effects on others. This interconnectedness requires holistic consideration and a reshaping of sustainability efforts to achieve comprehensive and balanced solutions.

(4) The three types of sustainability evaluation methodologies identified by airports are also widely used in other industries; however, a widely accepted consensus among

researchers asserts that *"no single approach suffices for all contexts"*, thus advocating for the use of a combination of methodologies in sustainability evaluation.

Based on the meta-review, DEA and its extensions, Hybrid-MCDM, and composite index-based assessment are also widely used in industries beyond airports [1,5–8,31,32]. Of these, MCDM emerges as the most frequently deployed approach. Concurrently, other techniques such as Life Cycle Analysis (LCA) [6,7,31], Social Life Cycle Analysis (sLCA) [7], CBA [7,8], and Input–Output Analysis (I-O) [31] have also been utilized in sustainability evaluations.

Nevertheless, Ravetz [50] acknowledges the inherent challenges in achieving a flawless sustainability performance evaluation in our dynamic world, characterized by rapid change, interdependency, and uncertainty. Similarly, Gasparatos et al. [8] propose that no single tool can fully encapsulate the wide range of perspectives among stakeholders, advocating for a contextually driven selection from a variety of tools. In line with this sentiment, Walzberg et al. [31] recommend integrating methods across different disciplines. Bueno et al. [7] further contribute to the discussion, emphasizing the considerable flexibility of the MCDM approach. This adaptable nature of MCDM makes it particularly suitable for amalgamation with other methods. The consensus among these scholars highlights the necessity for a multifaceted approach in sustainability evaluation, given the complexities and varied contexts of sustainability issues. Our findings in Section 3.2 further support this consensus and demonstrate that it persists in the airport sector.

(5)  Alongside the recognized issue of comprehensiveness, the challenges of trade-offs and compensatory effects further compound the limitations in current sustainability evaluation methodologies.

Composite indicators and, under certain contexts, MCDM can introduce trade-offs and compensatory effects due to their additive nature [1,6,7,32]. Such additive processes can lead to distortions in intended outcomes [51], as important information may become obscured during aggregation. However, not all MCDM methods exhibit this issue, with some, such as ELECTRE, employing a non-compensatory approach [52]. Conversely, the compensatory issue in DEA arises from its linear programming nature rather than an additive one. It allows for overperformance in one dimension to compensate for underperformance in another, potentially marking a unit as efficient overall despite poor performance in certain dimensions. Apart from the inherent methodological constraints, the trade-off and compensatory issues are also accentuated by an inadequate understanding of the intricate interconnections and dependencies among sustainability components, further complicating the sustainability evaluation process.

(6)  A consensus has emerged among authors endorsing systems thinking as the future direction for sustainability evaluation.

The most recent literature consistently posits systems thinking as the future trajectory for sustainability evaluation [6,31]. Systems thinking, a perspective that considers interconnectedness, relationships, and contexts, aligns closely with sustainability's foundational principles [53]. It supports the understanding of the dynamic behaviour within the systems under study, facilitating more integrated and resilient sustainability solutions [54]. By embracing the complexity of interdependencies and feedback loops within the system, systems thinking can alleviate issues related to trade-offs and compensatory effects prevalent in traditional approaches. Despite Janic's [35] early recognition of the necessity to incorporate the *"airport as a system"* concept in sustainability evaluation, its practical application within the airport sector remains limited. System Dynamics (SD) and Agent-Based Modelling (ABM) are methodologies frequently employed to operationalize systems thinking in sustainability evaluation [31]. However, a significant challenge remains to objectively and comprehensively understanding the system. To this end, Walzberg et al. [31] advocate for integrating MCDM with systems thinking methodologies. On the one hand, MCDM's ability to handle multiple criteria simultaneously complements the holistic view of systems thinking; on the other hand, MCDM can accommodate diverse stakeholder perspectives

enriches the decision-making process, providing a broader understanding of complex sustainability issues.

## 5. Conclusions

This paper endeavors to address a significant gap in the academic literature concerning airport sustainability evaluations, despite the industry's marked efforts to pursue sustainable practices. The study extensively reviewed 33 academic articles dedicated to airport sustainability evaluation, placing special emphasis on 16 papers that employed specific methodologies to assess airport sustainability performance. The review indicated that the methodologies employed across these papers could be primarily classified into three categories: DEA and its extensions, Hybrid MCDM, and composite index-based assessment.

The Fuzzy Dynamic Network DEA emerged as a robust tool in the DEA category for airport sustainability evaluation. However, its effectiveness could be further enhanced by minimizing subjectivity and embedding feedback loops and causal relationships within the sustainability framework. In contrast, while Hybrid-MCDM proved to be a helpful supplementary tool, it lacks the depth to serve as a standalone methodology. The SMM, although providing an advanced perspective in contrast to the composite indicator approach, necessitates greater involvement from experts and stakeholders in establishing a comprehensive, systems-oriented view.

To ensure a broad perspective, the study also extended its scope beyond the airport sector, conducting a meta-review of sustainability evaluation methodologies in other sectors. This comparative analysis revealed a common challenge across sectors: the lack of universally adaptable, well-defined sustainability frameworks, as most studies tend to develop unique approaches and criteria. Moreover, the analysis exposed an overemphasis on assessment, which has sidelined the essential role of sustainability accounting in the overall evaluation process. Interestingly, the three types of methodologies identified within the airport sector were also prevalent in other industries, demonstrating a potential cross-industry standardization.

The meta-review further unveiled emerging trends and insights, such as a growing critique of the traditional understanding of sustainability, which is typically divided into three or four dimensions. There is an increasing consensus among researchers that a single approach is insufficient for all contexts, advocating for a blend of methodologies. Furthermore, alongside the well-recognized challenge of achieving comprehensiveness in sustainability evaluation, issues relating to trade-offs and compensatory effects were identified as additional complications.

Our findings indicate a necessary change in research focus, from solely evaluating airport sustainability to modeling it. Modeling, backed by systems thinking, allows a better grasp of the complex interactions and feedback loops within sustainability systems. Notably, this shift also provides a pathway for addressing the challenges of trade-offs and compensatory effects that are inherent in current sustainability evaluation methodologies. Moreover, the integration of systems thinking with MCDM offers a promising avenue for future research, potentially enriching the methodologies available for sustainability evaluation and management.

**Author Contributions:** Conceptualization, Data curation, Methodology, Formal analysis, Writing—original draft, X.J.; Funding acquisition, Project administration, Supervision, Validation, Writing—review & editing, R.M.; Supervision, Validation, Writing—review & editing, S.B. All authors have read and agreed to the published version of the manuscript.

**Funding:** The authors would like to thank the European Union project TULIPS (Grant agreement ID: 101036996) for funding this research.

**Institutional Review Board Statement:** Not applicable.

**Informed Consent Statement:** Not applicable.

**Conflicts of Interest:** The authors declare no conflict of interest.

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
