# Peer review of "Expanding Horizons: A Review of Sustainability Evaluation Methodologies in the Airport Sector and Beyond"

_sustainability, doi:10.3390/su151511584_

Round 1
Reviewer 1 Report
Dear authors, I congratulate you for the interesting review prepared on an object of great relevance to the environmental theme. I have just a few considerations, set out below:
1. Table 2 should be presented in a clearer and more structured way. I believe that presenting in landscape mode offers greater clarity.
2. Author Janic is inappropriately cited in the paragraph preceding Table 2 (there is no line numbering in the text), as the citation omits the year of the study.
3. Perhaps the most important observation refers to the proposal to organize the text divided into sections, as proposed in the introduction, but throughout it, I observed that there was not due compliance, making it a little confusing.
Author Response
We deeply appreciate the opportunity afforded to us to revise and refine our manuscript, and to offer additional clarity where required.
Regrettably, there appears to have been a misalignment between the version of the manuscript we submitted and the version that was subsequently reviewed. In adherence to the journal's submission guidelines, we had understood that free format submissions were permissible, and thus, we used our own distinct template for the manuscript. However, it seems that post-submission, our manuscript underwent a reformatting process to align with the journal's traditional template, resulting in substantial changes to the document layout. The headings and tables were particularly impacted, leading to a presentation that did not reflect our original organization and coherence.
In our resubmission, we have taken decisive steps to rectify these issues. Specifically, we have provided two distinct versions of the manuscript. The first is our original manuscript, in which we employed the 'Track Changes' feature of Microsoft Word to transparently display all modifications made in response to the reviewers' feedback. We believe this version effectively highlights the changes undertaken.
The second version is a reformatted manuscript, duly aligned with the journal's template. Although the 'Track Changes' are not visible in this version, it ensures a clear and structurally sound representation of our work that adheres to the journal's formatting standards.
Reviewer 2 Report
First of all, I appreciate the opportunity to review the paper Expanding horizons: A review of sustainability evaluation methodologies in the airport sector and beyond. The paper deals with very actual problems. However, the paper has serious shortcomings. This paper is not for review! The preliminary check was not done!
· The paper is not written in the manner of scientific paper writing.
· The paper is not formatted and prepared in accordance with the template.
· All subsections are numbered as 1.1 (For example 1.1 Sustainability and Sustainability evaluation; 1.1. Airport sustainability evaluation; 1.1. Descriptive Analysis, etc.)
· On the other side very important sections are not numbered (for example: Methods, Materials and Content).
· The review paper should not just be a list of what everyone has done but should identify trends and gaps in the literature and offer suggestions for furthering the field relative to the specific phenomenon, with a VERY STRONG CRITICAL VIEW AND VERY STRONG METHODOLOGY.
· It is necessary to understand the purpose and aim of the paper as well as its "position" in relation to previous research (also gap analysis).
· The results are not reliable.
· The separate section Practical and theoretical implications (or Discussion) is missing. The existing section. This confirms the lack of scientific and practical contributions.
· The paper has no scientific and practical contributions.
.
Author Response
Thank you for taking the time to provide your feedback on our paper, "Expanding horizons: A review of sustainability evaluation methodologies in the airport sector and beyond." We appreciate your candid review and accept that there are areas where improvements can be made.
However, we would like to clarify an unfortunate situation that we believe may have led to some of the issues you pointed out. The version of the manuscript reviewed by you appears to be different from the one we initially submitted. We followed the journal's submission guidelines, which mentioned that free format submissions are now acceptable, and thus, we used our own template for the manuscript. However, it appears our manuscript was reformatted to fit the journal's template, causing substantial changes to the layout of the document. The reshuffling has affected the numbering of subsections and the overall organization of the manuscript, leading to a presentation that may seem disorganized.
That being said, we have taken your comments seriously and would like to address them:
With regards to the "scientific manner" of writing and formatting in accordance with the template, we assure you that our original submission adhered to rigorous scientific writing standards. The disorganization due to reformatting issues unfortunately does not reflect the quality of the original manuscript.
On your point about identifying trends and gaps in literature, we would like to highlight that we have made an extensive effort to do so in the original manuscript. However, this might not be evident in the current version due to formatting issues. In our resubmitted version, we have made sure these sections are clearer and more explicit.
We understand your concern about the reliability of results, the practical and theoretical implications, and the paper's scientific and practical contributions. Rest assured, in our original manuscript, we took great care to ensure the integrity of our findings and their implications for both the academic and practical fields. We will also make sure these points are clearly communicated in our revised manuscript.
In light of the above, we kindly ask that you consider reviewing our revised version where we have addressed these issues and highlighted the changes in response to your valuable comments.
Thank you once again for your feedback, and we look forward to hearing from you.
Reviewer 3 Report
This is a well-written meta-analysis of (airport) sustainability evaluations, which a methodological focus.
The paper starts with a good introduction into the concepts of sustainability accounting and assessment as part of sustainability evaluation, followed by a clear reasoning for the relevance of airport sustainability evaluations, and with a clear introduction of the objectives and methodological approach. The paper contains a comprehensive and well written literature evaluation and classification according to key methodologies, and a critical assessment of such methodologies citing key literature. Derivation of sound conclusions, albeit without a graphical overview, e.g. in a table.
I only have minor comments:
Abstract, 1st sentence: As the reviews goes beyond the airport sector, this sentence should be rephrased, e.g. "sustainability evaluations with a focus on the airport sector"
The limitations mentioned on page 2 should be explained in more detail, with actual references for each item. Maybe use a table.
The numbering in the introduction is wrong. The numbering from section 5 onwards also seems to be wrong (“conclusions” 6-12).
1.1, 1st para add the "enabling" (of worldwide trade) aspect as reasoning for economic growth
Page 3, 1st paragraph: Is there any proof for the claim that sustainability commitment can help attracting passengers? If not, remove or re-phrase, as – otherwise – this would be just speculation.
Please explain how you have allocated the papers to the 4 categories? What about papers that would have fit to more than just one group?
Should the third category of papers not better be called "Airport building sustainability performance studies"?
Upham/Mills 2005 also lack the social and economic dimension, right? (Page 6)
Table 2 - what is meant by the operational dimension? Please explain as one could also argue that all kinds of sustainability effects stem from operations (+construction), with operations not a sustainability dimension of its own.
Why did you select Adler at al 2013 when you acknowledge that the paper actually is not about sustainability? Same for Wang/Song 2020?
Conclusion 9 (DEA etc. also used in other industries): What is the message of this conclusion? Wouldn’t it be more relevant to assess if there is a “best” methodology, or for which questions which methodology should be applied (which translates to Conclusion #10)?
Structure: Pages 12-14 contain conclusions – before the actual conclusion section on pp14ff. Please re-structure, or rename the last section "summary".
mostly ok
Author Response
Dear Reviewer,
We would like to express our gratitude for the opportunity to revise our manuscript and provide further clarity on a few matters.
Before delving into the content-specific questions, we feel it's important to address a significant formatting issue. The version of the manuscript reviewed by the academic editor and reviewers, unfortunately, does not match the version we initially submitted.
Following the journal's submission guidelines, we understood that free format submissions are now accepted, and accordingly, we used our own template for the manuscript. However, it appears our submission was reformatted to fit the journal's template, causing considerable alterations to the layout of the document. The restructuring particularly affected headings and tables, resulting in a disorganized presentation.
We have taken measures to clarify these issues in our resubmission. Specifically, we have included two versions of the manuscript. The first version is our original manuscript where we have used the 'Track Changes' function in Microsoft Word to clearly indicate all modifications made in response to the reviewers' comments. We believe this version will provide the clearest indication of the changes made.
The second version is a revised manuscript formatted according to the journal's template. In this version, while the 'Track Changes' are not visible, it should provide a clear and organized presentation of our work that aligns with the journal's formatting requirements.
As for the content-related suggestions and inquiries made by the reviewers, we have made every effort to address each one thoroughly and comprehensively. We have outlined our responses to each point in the following section:
Comment: Abstract, 1st sentence: As the reviews goes beyond the airport sector, this sentence should be rephrased, e.g. "sustainability evaluations with a focus on the airport sector"
Reply: Thank you for your insightful feedback on our abstract. We agree with your suggestion to highlight our study's dual focus on sustainability evaluations within the airport sector as well as across a range of other industries. We have revised the abstract accordingly to reflect this perspective right from the outset.
Comment: The limitations mentioned on page 2 should be explained in more detail, with actual references for each item. Maybe use a table.
Reply: Thank you for your suggestions regarding the need for more detailed explanation of the limitations with appropriate references. In response to your suggestion, I have revised the paragraph to explicitly list and describe each limitation individually. For each limitation, I've provided relevant references to support the statements made.
Comment: 1.1, 1st para add the "enabling" (of worldwide trade) aspect as reasoning for economic growth
Reply: Thank you for your insightful comment on emphasizing the role of airports in enabling worldwide trade, a significant driver of economic growth. I appreciate the importance of highlighting this aspect in the context of sustainable development within the aviation industry.
In response to your comment, I have revised the relevant sentence in the paragraph to read: "These benefits encompass economic growth from enabling worldwide trade, enhanced connectivity, and the creation of job opportunities, among others."
Comment: Page 3, 1st paragraph: Is there any proof for the claim that sustainability commitment can help attracting passengers? If not, remove or re-phrase, as – otherwise – this would be just speculation.
Should the third category of papers not better be called "Airport building sustainability performance studies"?
Reply: Thank you for your insightful comment regarding the need for evidence to support the claim that a commitment to sustainability can attract passengers, airlines, and businesses. I have found two references from ACI which align with this assertion. Accordingly, I have revised the sentence to include these references, providing the necessary proof to support the claim.
Comment: Please explain how you have allocated the papers to the 4 categories? What about papers that would have fit to more than just one group?
Reply: I would like to address your query about the allocation of papers to the four distinct categories we used. Upon a thorough review and categorization process, we assure that the classification we've made ensures there's no overlap between the categories, each paper fitting exclusively into one group.
Airport sustainability performance evaluation studies: These focus specifically on the operational phase and use performance indicators to quantify an airport's overall sustainability. Studies under this category do not examine specific strategies or thematic areas, thus differentiating them from the other categories.
Airport sustainability strategy/measure/project evaluation studies: These analyze the impact of a specific strategy, measure, or project on an airport's sustainability. Unlike studies evaluating overall performance, these concentrate on a particular initiative or project, providing a focused impact assessment.
Airport terminal sustainability evaluation studies: These studies focus on all phases of an airport terminal's lifecycle - from design and construction to operation and maintenance - with the aim to ensure sustainability throughout. This focus on a specific physical structure within the airport terminal sets them apart from the broader or strategy-specific evaluations of the other categories.
(Additionally, we have updated the name of the third category from "Airport Sustainability Building Performance Studies" to "Airport Terminal Sustainability evaluation Studies." We believe this change better encapsulates the scope of this category, emphasizing its focus on the sustainability of airport terminal buildings.)
Mono-thematic airport sustainability performance evaluation studies: These papers differentiate themselves by addressing sustainability performance within specific themes, such as water or waste management systems. Their thematic focus sets them apart from the broader or project-specific evaluations of the other categories.
We hope this clarifies your concerns. We appreciate your insightful comments that have led to improvements in our manuscript.
Comment: Upham/Mills 2005 also lack the social and economic dimension, right? (Page 6)
Reply: Yes, in the text we mentioned “According to both of Upham’s articles, airport sustainability encompasses two dimensions: environmental and operational.”
Comment: Table 2 - what is meant by the operational dimension? Please explain as one could also argue that all kinds of sustainability effects stem from operations (+construction), with operations not a sustainability dimension of its own.
Reply: Thank you for your comment, which provides us an opportunity to clarify the 'Operational' dimension of our review framework. In the indicators review process, we define the 'Operational' dimension as relating specifically to the operational performance and efficiency of airports, which includes factors such as aircraft movement, passenger or cargo throughput, etc. We acknowledge that operations inherently have economic, environmental, and social impacts; however, in our framework, these impacts are accounted for under the respective dimensions. The 'Operational' dimension is therefore distinct, focusing on the internal operational processes and efficiency of airports themselves. It is not about the sustainability impacts of these operations (which are captured in the other dimensions), but rather the operational performance and effectiveness of the airport in its role as a transport service provider. We have also marked this under the table as a further explanation.
Comment: Why did you select Adler at al 2013 when you acknowledge that the paper actually is not about sustainability? Same for Wang/Song 2020?
Reply: Thank you for your valuable comments and queries regarding the inclusion of the studies by Adler et al. (2013) and Wang and Song (2020) in our analysis. We appreciate this opportunity to elaborate on our rationale.
The key criterion we applied when selecting studies for our review was the self-proclaimed focus of each paper. Both Adler et al. (2013) and Wang and Song (2020) titled their research as assessment into airport sustainability, which aligns with the thematic focus of our review.
Despite the fact that Adler et al. (2013) used measures that are conventionally associated with operational efficiency, their declared intent was to study sustainability performance. The authors attribute this to the financial and operational constraints faced by small regional airports, which can make achieving sustainability a challenge. Similarly, while Wang and Song (2020) didn't fully employ a 'network DEA' approach as claimed, their intention to evaluate sustainability performance is explicit.
The inclusion of these studies in our review serves to illustrate the varying interpretations and applications of sustainability evaluation within the existing literature. Moreover, it underlines the need for a more standardized approach to such evaluations and provides important lessons for future research in this area.
Comment: Conclusion 9 (DEA etc. also used in other industries): What is the message of this conclusion? Wouldn’t it be more relevant to assess if there is a “best” methodology, or for which questions which methodology should be applied (which translates to Conclusion #10)?
Reply: Thank you for your insightful comment regarding the conclusions drawn from our analysis of sustainability evaluation methodologies. Upon reflection, we agree that the two points we initially presented separately, concerning the widespread usage of certain evaluation methodologies and the scholarly consensus on the insufficiency of a single approach, could be more effectively synthesized into a singular, holistic insight.
In response to your suggestion, we have revised our manuscript to combine these two points.

Reviewer 4 Report
1) The Authors mentioned 3 main types of methodologies DEA, MCDM, and composite index-based assessments. Please explain why those types were distinguished. Are there any other significant types?
2) The Authors give definitions of evaluation and assessment in the second paragraph of page 2. While they are correct, it is worth mentioning that researchers probably do not always base on such definitions. Sometimes, assessment and evaluation can be considered different or even synonyms.
3) At the end of page 11, the Authors comment that in the paper by Yangmin et al. not enough attention was paid to several things. Why do the Authors think those parameters are worth paying more attention to?
Formal issues:
- Figure 1 and its caption are separated into 2 pages.
- Figures 2 and 3 are separated from their captions.
- Table 2 divides the main text in the middle of a sentence.
- Point 1.1 is used twice: on page 2 and page 10.
- Different font is used in point 12 (Page 14).
Author Response
Dear Reviewer,
We would like to express our gratitude for the opportunity to revise our manuscript and provide further clarity on a few matters.
Before delving into the content-specific questions, we feel it's important to address a significant formatting issue. The version of the manuscript reviewed by the academic editor and reviewers, unfortunately, does not match the version we initially submitted.
Following the journal's submission guidelines, we understood that free format submissions are now accepted, and accordingly, we used our own template for the manuscript. However, it appears our submission was reformatted to fit the journal's template, causing considerable alterations to the layout of the document. The restructuring particularly affected headings and tables, resulting in a disorganized presentation.
We have taken measures to clarify these issues in our resubmission. Specifically, we have included two versions of the manuscript. The first version is our original manuscript where we have used the 'Track Changes' function in Microsoft Word to clearly indicate all modifications made in response to the reviewers' comments. We believe this version will provide the clearest indication of the changes made.
The second version is a revised manuscript formatted according to the journal's template. In this version, while the 'Track Changes' are not visible, it should provide a clear and organized presentation of our work that aligns with the journal's formatting requirements.
As for the content-related suggestions and inquiries made by the reviewers, we have made every effort to address each one thoroughly and comprehensively. We have outlined our responses to each point in the following section:
Comment: 1) The Authors mentioned 3 main types of methodologies DEA, MCDM, and composite index-based assessments. Please explain why those types were distinguished. Are there any other significant types?
Reply: Thank you for your comment about our categorization of methodologies. In Figure 3, we classified methodologies from the 16 airport sustainability studies into three types: DEA, MCDM, and composite index-based assessments. Our use of the term "broadly" in the original manuscript may have suggested we overlooked other methodologies, which is not the case. To remove any ambiguity, we've now eliminated this term from our manuscript. All methodologies from the reviewed studies fit into the mentioned three types. We appreciate your attention to this detail and trust this revision clarifies our methodological classification.
Comment: 2) The Authors give definitions of evaluation and assessment in the second paragraph of page 2. While they are correct, it is worth mentioning that researchers probably do not always base on such definitions. Sometimes, assessment and evaluation can be considered different or even synonyms.
Reply: Thank you for your valuable feedback and your note on the usage of the terms "evaluation" and "assessment". We acknowledge that these terms can be used differently across various disciplines, and sometimes interchangeably.
In response to your comment, we have revised the related paragraph in our manuscript to clarify our definitions of "evaluation" and "assessment" based on Scriven (1991) and Büyüközkan and Karabulut (2018). We have also emphasized that these definitions are not universally accepted, but they provide a meaningful distinction for the purposes of our study.
We believe these changes provide greater clarity and address your concern. We appreciate your attention to this detail, which has helped improve the precision and rigor of our paper.
Comment: 3) At the end of page 11, the Authors comment that in the paper by Yangmin et al. not enough attention was paid to several things. Why do the Authors think those parameters are worth paying more attention to?
Reply: Thank you for your comment. We concur with the point you made about the role of indicators as vital components of the system. Even though Yangmin et al. (2021) claim their focus is on the system as a whole, we believe that a thorough scrutiny of the individual assessment indicators, including their validity, correlation, and robustness, is fundamental for a comprehensive evaluation of the system itself. The choice of indicators directly affects the interpretation and composition of the system. Moreover, a sensitivity analysis would be beneficial to understand how variations in these indicators can impact the overall system. The original authors themselves recognize the need for further examination of their chosen AIC indicators, corroborating our standpoint. We have incorporated these perspectives into our critique in the manuscript.

Round 2
Reviewer 2 Report
Unfortunately, the only change is the technical change. Is this possible “The version of the manuscript reviewed by you appears to be different from the one we initially submitted”?!
The authors’ response and revised manuscript are focused on technical changes, but not on the main substantive objections.
However, all material shortcomings are still present.
· The review paper should not just be a list of what everyone has done, but should identify trends and gaps in the literature and offer suggestions for furthering the field relative to the specific phenomenon, with a VERY STRONG CRITICAL VIEW AND VERY STRONG METHODOLOGY.
· It is necessary to understand the purpose and aim of the paper as well as its "position" in relation to previous research (also gap analysis).
· The results are not reliable.
· The separate section Practical and theoretical implications (or Discussion) is missing. The existing section. This confirms the lack of scientific and practical contributions.
· The paper has no scientific and practical contributions.
.
Author Response
- The review paper should not just be a list of what everyone has done, but should identify trends and gaps in the literature and offer suggestions for furthering the field relative to the specific phenomenon, with a VERY STRONG CRITICAL VIEW AND VERY STRONG METHODOLOGY.
Reply: We concur that the emphasis of a review paper should extend beyond summarising the existing literature to identifying trends, highlighting gaps, and proposing recommendations. This approach is reflective of the three research questions we posed in the Introduction section, which we believe have been thoroughly addressed in our manuscript.
In Section 3.2, 'Assessment Methodologies', we have categorized and critically examined the methodologies employed in existing airport sustainability performance evaluation studies. Our discussion extends beyond simple categorization to explore the limitations inherent to each methodology, enhancing comprehension through the use of bullet points for better presentation.
Further, in Section 4, 'Comparative Discussion beyond Airport Sector', we conducted an extensive comparison of sustainability evaluation methodologies both within and beyond the airport sector. This comparison led to the revelation of six key findings, each presented in bullet points supported by evidence, thereby addressing similarities, differences, and potential future trends.
In terms of methodology, we utilized a systematic literature review for papers within the airport sector, complemented by a meta-review for studies outside the airport sector. We believe this robust approach adequately addresses our research questions and enhances the validity of our conclusions.
- It is necessary to understand the purpose and aim of the paper as well as its "position" in relation to previous research (also gap analysis).
Reply: Our paper is a review of airport sustainability evaluation studies with a distinct focus on evaluation methodologies. We have endeavored to situate our study within the body of existing literature and have articulated a clear "position" by conducting an in-depth analysis of the prevailing gaps. Specifically, we have analyzed the disparities among three types of airport sustainability evaluation studies, as well as differences in methodologies utilized in the airport sector and beyond. We would appreciate further clarification if there are any specific areas where you feel our gap analysis is lacking.
- The results are not reliable.
Reply: We have meticulously sourced our papers through recognized databases such as Scopus and WoS. We have carefully screened, read, and analyzed these papers to ensure the reliability and validity of our findings. We invite you to provide specific examples or areas where you question the reliability, allowing us to address your concerns more effectively.
- The separate section Practical and theoretical implications (or Discussion) is missing. The existing section. This confirms the lack of scientific and practical contributions. The paper has no scientific and practical contributions.
Reply: We appreciate your viewpoint but respectfully disagree. Our paper provides both theoretical and practical contributions by offering a comprehensive review of methodologies used in airport sustainability evaluations, a topic that remains underexplored in the current literature. We have identified gaps and highlighted the need for a paradigm shift from evaluation to modeling sustainability, which has significant implications for the research community and the industry. Our work also advocates for an integrated approach, combining systems thinking with MCDM, which holds promising potential for future research and applications in sustainability evaluation.

Round 3
Reviewer 2 Report
Your paper is again out of the template.
.